# Electrochemical Biosensors Driving Model Transformation for Food Testing

**DOI:** 10.3390/foods14152669

**Published:** 2025-07-29

**Authors:** Xinxin Wu, Zhecong Yuan, Shujie Gao, Xinai Zhang, Hany S. El-Mesery, Wenjie Lu, Xiaoli Dai, Rongjin Xu

**Affiliations:** 1School of Food and Biological Engineering, Jiangsu University, Zhenjiang 212013, China; 19825549348@163.com (X.W.); 2212418018@stmail.ujs.edu.cn (Z.Y.); 2212418006@stmail.ujs.edu.cn (S.G.); 2School of Energy and Power Engineering, Jiangsu University, Zhenjiang 212013, China; elmesiry@ujs.edu.cn (H.S.E.-M.); dai_xiaoli1980@126.com (X.D.); xurongjin@ujs.edu.cn (R.X.)

**Keywords:** electrochemical biosensor, nanostructures, specific capture, food safety, food quality

## Abstract

Electrochemical biosensors are revolutionizing food testing by addressing critical limitations of conventional strategies that suffer from cost, complexity, and field-deployment challenges. Emerging fluorescence and Raman techniques, while promising, face intrinsic drawbacks like photobleaching and matrix interference in opaque or heterogeneous samples. In contrast, electrochemical biosensors leverage electrical signals to bypass optical constraints, enabling rapid, cost-effective, and pretreatment-free analysis of turbid food matrices. This review highlights their operational mechanisms, emphasizing nano-enhanced signal amplification (e.g., Au nanoparticles and graphene) and biorecognition elements (antibodies, aptamers, and molecularly imprinted polymers) for ultrasensitive assay of contaminants, additives, and adulterants. By integrating portability, scalability, and real-time capabilities, electrochemical biosensors align with global food safety regulations and sustainability goals. Challenges in standardization, multiplexed analysis, and long-term stability are discussed, alongside future directions toward AI-driven analytics, biodegradable sensors, and blockchain-enabled traceability, ultimately fostering precision-driven, next-generation food safety and quality testing.

## 1. Introduction

The high-performance inspection of food supply is critical to safeguarding public health and ensuring regulatory compliance [1,2,3]. Regarding food supply, food safety and quality assurance are critical issues that have attracted enormous attention around the world [4,5,6,7]. Generally, food safety is mentioned in relation to contaminant evaluation, including heavy metals, pesticide residues, mycotoxins, and pathogens, aligning with global food safety testing regulations and standards [8,9,10,11,12]. Simultaneously, quantifying additive analysis (sugar levels, antioxidants, and additives) ensures adherence to permissible thresholds [13,14,15]. Moreover, monitoring adulterant species (e.g., melamine in dairy products) combats food fraud and protects economic equity [16,17,18,19,20]. These challenges demand rapid, scalable assay tools to ensure public health.

Thus far, food safety and quality testing rely on conventional and emerging methodologies [21,22,23,24,25,26]. The conventional patterns, such as high-performance liquid chromatography (HPLC) [27,28,29] and gas chromatography–mass spectrometry (GC-MS) [30,31,32], remain foundational for detecting various species due to their high sensitivity and ability to handle complex matrices [33,34,35]. However, these techniques often require expensive instruments, laborious sample treatment, and specialized personnel, restricting their application in field or resource-scarce settings. The emerging patterns, including visible near-infrared spectroscopy [36,37,38], colorimetric sensing [39,40,41,42], fluorescence sensing [43,44,45], and Raman spectroscopy [46,47,48,49], have gained traction in food safety and quality assurance. Nevertheless, during their practical adoption, the mentioned spectroscopic approaches frequently face unresolved challenges from the characteristics of the light source itself [50,51,52]. To be specific, fluorescence-based platforms suffer from photobleaching and matrix interference in opaque samples like dairy or meat [53,54]. Surface-enhanced Raman spectroscopy (SERS) [55,56], though highly specific for chemical fingerprinting, is suitable for quantitation in heterogeneous samples. Unlike fluorescence/Raman methods, electrochemical biosensing bypasses these constraints by leveraging electrical signals [57,58,59], free of matrix interference from turbidity and color, enabling analysis of turbid food matrices without pretreatment [60,61,62].

Considering the desirability of food analysis in safeguarding public health and ensuring regulatory compliance [63,64,65], this review systematically explores the application of electrochemical sensing technologies in food safety and quality monitoring [66]. Herein, the operational mechanism of electrochemical biosensors was discussed, alongside nano-enhanced signal amplification and biorecognition elements (Figure 1). Moreover, the application across food matrices spans the evaluation of low levels of contaminants and additive analysis alongside adulterant species. Meanwhile, the challenges and future directions of electrochemical biosensing toward food testing are also discussed, aiming for precision, scalability, and sustainability in the next-generation food safety ecosystem. We have created a graphical summary of the entire article to facilitate readers’ comprehension (Table 1).

## 2. Electrochemical Sensing Principles

### 2.1. Mechanisms of Signal Output

Electrochemical signals originate from redox reactions between target analytes and the electrode surface, modulated by nanostructure-engineered interfaces [67]. When analytes (e.g., pesticides and pathogens) interact with recognition elements (aptamers and antibodies) immobilized on the sensing interfaces, electron transfer kinetics are altered. As for signal output, the analytes themselves (e.g., Pb^2+^, Cu^2+^, and vitamin C) [68,69,70,71] with redox activity could be directly detected to obtain electrochemical signals [72]. On the other hand, signal probes based on enzymes (e.g., horseradish peroxidase and laccase) are leveraged to detect analytes by coupling catalytic reactions [73,74]. For example, glucose oxidase oxidizes glucose to gluconic acid, releasing electrons measurable as current response [75,76]. Moreover, nanostructures with enzyme-like properties, such as metal–organic frameworks (MOFs) [77,78,79] and metallic oxides [80,81], can be used as biomimetic catalysts (named as nanozymes) for signal output. Furthermore, redox mediators (e.g., ferrocene derivatives [82], methylene blue [69], and thionine) are also exploited to design signal probes for generating current response.

### 2.2. Signal Acquisition Modalities

Modern electrochemical systems take advantage of multifunctional readout strategies tailored to food matrices [83,84,85], consisting of direct current (DC), alternating current (AC), and hybrid and emerging modalities.

(1) Direct current techniques in food analysis

Direct current (DC) techniques include voltammetry and amperometry, in which voltammetric methods (e.g., differential pulse voltammetry, DPV; square-wave voltammetry, SWV; linear sweep voltammetry, LSV) modulate electrode potentials to induce redox reactions, measuring resultant current–voltage profiles. For instance, LSV was applied to measure sunset yellow in soft drinks [86]; SWV peak current increases gradually with the increasing Hg^2+^ concentration in dairy products via the potential window from −0.50 to 0 V (vs. SCE) under a step potential of 4 mV [87]. DPV achieved low-level heavy metals or hydroxyl-sanshools [88,89]. Moreover, the amperometry model is exploited to monitor steady-state currents from catalytic reactions, exemplified by glucose oxidase-based biosensors [90]. Recent advances include nanochannel-confined DPV (V-doped Co_3_O_4_/g-C_3_N_4_ heterostructure) for simultaneous electrochemical detection of ascorbic acid/dopamine. The chronoamperometric sensor achieves user-friendly and low-cost tyramine tracking during the spoilage of fish samples [91] (Figure 2).

(2) Alternating Current techniques in food analysis

Alternating current (AC) techniques include electrochemical impedance spectroscopy (EIS) and conductometric sensors [98,99,100] (Figure 3), in which EIS analyzes frequency-dependent impedance (Z) to probe interfacial phenomena. The EIS sensor classifies the freshness of fish or carp samples from different origins [100,101,102,103]. The EIS platform assesses the quality of Atlantic salmon/rainbow trout, chilled/frozen–thawed salmon, and fresh/stale salmon via charge-transfer resistance (Rct) shifts [104,105]. A conductometric sensor is used for quantification of arginine in dietary supplements.

(3) Hybrid and emerging modalities in food analysis

Hybrid and emerging modalities include photoelectrochemistry (PEC) and self-powered systems [110] (Figure 4), in which PEC sensors integrate light excitation with electron transfer [111], exemplified by CdTe/WS_2_ heterojunctions detecting zearalenone in cereal crops based on photoinduced electron transfer [112]. Self-powered systems harvest energy from analyte reactions, such as electrochemical aptasensing platforms enabling ultrasensitive and real-time detection of microcystin-RR [113]. The ratiometric aptasensing toward aflatoxin B1 is achieved based on plasmon-modulated competition between photoelectrochemistry-driven and electrochemistry-driven redox of methylene blue [114].

### 2.3. Signal Amplification Strategies

Electrochemical signal amplification is pivotal for achieving ultra-sensitive assay of trace-level analytes [57,118,119], enabling food safety and quality assurance by overcoming inherent limitations of low analyte concentrations and background noise [110,120,121,122], thereby enhancing sensitivity down to attomolar levels; amplification mechanisms primarily exploit catalytic cascades (e.g., enzyme-linked redox reactions or nanozyme-mediated catalysis) [123,124,125] and molecular recognition-driven replication (e.g., DNA hybridization chain reactions or CRISPR-Cas-triggered nucleic acid amplification) [82,126,127]. The commonly integrated strategies include redox cycling systems (e.g., [Ru(NH_3_)_6_]^3+^/mediator pairs for signal multiplication), signal probes (e.g., enzyme–DNA conjugates) [128,129], and nanostructured sensing interfaces; they collectively advance real-time, portable, and cost-effective sensing platforms through the convergence of nanotechnology, molecular biology, and electrochemistry for transformative food testing applications [130] (Figure 5).

## 3. Nanostructure-Sensitized Electrochemical Assay

Owing to unique physicochemical properties, nanomaterials play a transformative role in electrochemical sensing by amplifying signals via high surface-to-volume ratios (e.g., graphene or MOFs maximizing active sites), accelerating electron transfer kinetics through conductive pathways like carbon nanotube “bridges”, enabling targeted recognition via surface functionalization such as aptamer-decorated nanoparticles, and suppressing interference by size-exclusion effects of nanoporous membranes; applied nanomaterials are broadly categorized as metallic nanoparticles (e.g., catalytic Au/Ag/Pt), carbon-based materials (e.g., graphene/CNTs/carbon dots), and porous frameworks (e.g., synergistically amplifying MOFs).

### 3.1. Metallic Nanomaterials for Sensitivity Enhancement

Metallic nanomaterials play a pivotal role in advancing electrochemical sensing technologies by significantly enhancing sensitivity, selectivity, and stability [135,136] (Figure 6A). These materials leverage their unique physicochemical properties, such as high electrical conductivity, catalytic activity, and tunable surface chemistry, to amplify signals and enable precise detection of trace analytes in complex systems [137]. Their ability to accelerate electron transfer kinetics, catalyze redox reactions, and maximize surface interactions makes them indispensable for applications ranging from medical diagnostics to environmental monitoring.

The sensitivity enhancement mechanisms of metallic nanomaterials are rooted in several key principles. Noble metals like gold (Au) and platinum (Pt) act as efficient electron mediators, reducing charge-transfer resistance at electrode interfaces. For instance, gold nanoparticles (AuNPs) serve as “electron highways” [138], bridging redox-active molecules and electrode surfaces to amplify signals. Transition metals and their oxides, such as iron oxide (FeO_2_) and manganese dioxide (MnO_2_), exhibit intrinsic catalytic properties that lower reaction overpotentials, thereby boosting currents for target analytes like tetracycline, bisphenol A, and catechol [139]. Nanostructuring techniques further enhance performance by creating porous or dendritic architectures that maximize active surface areas, as for glucose sensing [140]. Additionally, synergistic effects in bimetallic systems (e.g., Au@Pd core–shell structures) combine the catalytic prowess of one metal with the stability of another [141], enabling dual functionality in sensing toward AFB_1_ in complex matrices.

Metallic nanomaterials are categorized into noble metals and alloys, transition metal oxides, bimetallic core–shell structures, and metal chalcogenides. Noble metals like Au and Pt are widely used in non-enzymatic glucose sensors [142] and aptasensors, where AuNPs functionalized with aptamers achieve ultra-low detection limits for aflatoxin B1 in the realm of food safety monitoring. Transition metal oxides, such as Fe_3_O_4_ nanoparticles [11,143,144], could enhance differential pulse voltammetric signals in monitoring of vanillin in food products [145]. Bimetallic systems, such as Ag@Au core–shell nanoparticles [146], were applied for sensing toward single and simultaneous detection of AFB1 and OTA [147].

Looking ahead, the integration of sustainable synthesis methods—such as bioinspired nanoparticle fabrication—and hybrid material systems will further democratize metal-based electrochemical sensors. These advancements promise to expand applications in food safety and quality, cementing metal-based materials as cornerstone components in the next generation of high-sensitivity sensing technologies.

**Figure 6 foods-14-02669-f006:**
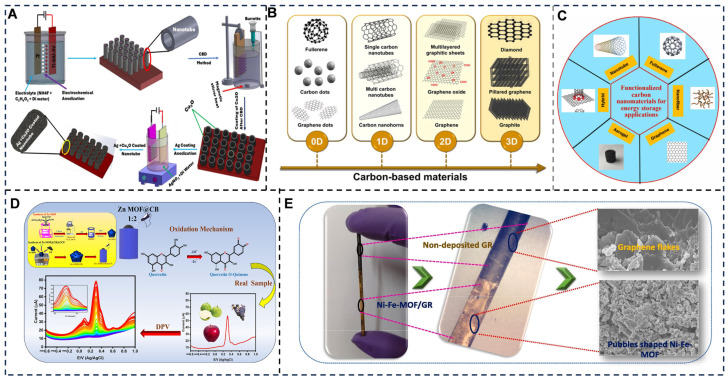
(**A**) Schematic illustrations of various stages of synthesis for developing Ag-Cu_2_O@TNTs nanohybrids [148]. (**B**) Summary of carbon-based materials of various dimensions [149]. (**C**) Different carbon nanomaterials are used for energy storage applications [150]. (**D**) Illustration of electrochemical determination of quercetin using Zn-MOF@CB nanocomposite modified GCE [151]. (**E**) Pictorial illustration of free-standing bimetallic metal–organic framework-modified graphite rod working electrode [152]. Adapted with permission from Refs. [148,149,151,152]. Kumar et al. (2024), Venkateswara Raju et al. (2023), Selvam et al. (2025) and Dey et al. (2025).

### 3.2. Carbon-Based Materials for Sensitivity Enhancement

Carbon-based materials (CBMs) are revolutionizing electrochemical sensing by significantly enhancing sensitivity through their unique structural and electronic properties [153,154] (Figure 6B,C). Their high surface area, exemplified by graphene and porous carbon, provides abundant active sites for analyte adsorption, directly lowering detection limits. Conductive pathways in materials, such as carbon nanotubes (CNTs) [155,156] and reduced graphene oxide (rGO) [157], accelerate electron transfer, reducing charge-transfer resistance and improving signal-to-noise ratios. Heteroatom doping (e.g., nitrogen or sulfur) further modulates electronic properties, boosting catalytic activity for redox reactions.

CBMs are classified into graphene derivatives, carbon nanotubes, carbon dots, porous carbons, and carbon composites. Graphene oxide (GO) and its reduced form (rGO) are widely used in glucose and heavy metal sensing [123,158,159], leveraging their tunable surface chemistry. Carbon nanotubes, both single-walled and multi-walled, excel in neurotransmitter monitoring and pathogen detection due to their high aspect ratio and functionalization versatility. Carbon dots [160], especially doped variants, enable selective detection of explosives. Porous carbons, such as ordered mesoporous carbon (OMC) [161], enhance pesticide residue analysis through analyte preconcentration [162].

Innovative applications include MXene–graphene hybrids for ultrasensitive detection of endosulfan [163]. Despite their promise, challenges remain in scalable synthesis of defect-free materials, long-term stability under harsh conditions, and cost-effective industrial deployment. Emerging trends focus on AI-driven design of tailored CBM–analyte interactions, self-powered sensors integrating triboelectric nanogenerators, and sustainable biomass-derived carbons for eco-friendly sensing.

By bridging nanotechnology and biotechnology, carbon-based materials are pushing the boundaries of sensitivity in electrochemical sensing, enabling breakthroughs to monitor food safety and quality. Their structural adaptability and multifunctionality continue to drive innovations, offering real-time, high-precision solutions across scientific and industrial domains [164].

### 3.3. Porous Frameworks for Sensitivity Enhancement

Porous framework materials (PFMs), including MOFs [165,166,167], covalent organic frameworks (COFs) [168], and porous organic polymers (POPs), have revolutionized electrochemical sensing by leveraging their ultrahigh surface areas, tunable porosity, and programmable functionalities [61,169] (Figure 6D,E). These materials enhance sensitivity through mechanisms such as analyte preconcentration, selective molecular recognition, and catalytic signal amplification. Their modular architectures enable precise integration of active sites—such as metal clusters, redox-active linkers, or functional groups—that interact specifically with target molecules, enabling detection limits down to attomolar concentrations in complex food samples. For instance, MOFs like ZIF-8 achieve large surface areas [24,170], allowing efficient adsorption of catechin or dopamine, while COFs with π-conjugated systems facilitate electron transfer in trace metals analysis [171].

The sensitivity enhancement of PFMs stems from their structural and chemical versatility. Nanopores in frameworks like UiO-66 or MIL-101 act as molecular sieves [172,173], selectively trapping analytes such as heavy metal ions (Hg^2+^ and Pb^2+^) through coordination or electrostatic interactions, thereby enriching their local concentrations at electrode surfaces. Simultaneously, redox-active metal nodes (e.g., Fe^3+^ in MIL-101) or organic linkers (e.g., porphyrins in COFs) [174] catalyze key electrochemical reactions, lowering overpotentials and amplifying currents. Hybrid designs further optimize performance: MOF–graphene composites combine the porosity of MOFs with the conductivity of graphene [175]. Stimuli-responsive PFMs, such as photoresponsive COFs [159], introduce dynamic control, where light irradiation triggers substantial potential for aniline adsorption and detection [176]. The sensitive potentiometric sensor exploiting ultrathin two-dimensional nanosheets of Mn metal–organic framework (2D Mn-MOF-NSs) was prepared to determine Mn(II) ion content with accuracy and precision [177].

Looking ahead, PFMs are poised to redefine electrochemical sensing through intelligent design and interdisciplinary innovation. Advances in adaptive frameworks, machine learning-driven synthesis, and biodegradable materials will expand their role in food testing. By harmonizing molecular precision with functional robustness, PFMs will continue to push the boundaries of sensitivity, selectivity, and real-world applicability, cementing their status as cornerstone materials in next-generation analytical science.

## 4. Recognition Elements in Electrochemical Sensing Toward Food Testing

Recognition elements are the cornerstone of electrochemical sensing systems, dictating selectivity, sensitivity, and specificity by enabling targeted interactions with analytes across food domains [178]. These elements span biological, synthetic, and hybrid categories, each exploiting unique molecular mechanisms to bind or catalyze reactions with target species. Biological recognition elements, such as aptamers [179], antibodies [180], cells, protein scaffolds, and molecularly imprinted polymers (MIPs) [181], for ultra-specific analyte capture.

### 4.1. Aptamer-Enhanced Selectivity in Electrochemical Sensing

Aptamers are single-stranded DNA/RNA oligonucleotides or peptide molecules selected via SELEX (Systematic Evolution of Ligands by Exponential Enrichment) to bind targets with high specificity and affinity [182,183,184] (Figure 7A). Their selectivity in electrochemical sensing stems from three key mechanisms: conformational switching, molecular recognition, and signal amplification [185]. Conformational switching refers to structural reorganization (e.g., folding/unfolding) upon target binding, which alters electron transfer efficiency between redox tags (e.g., methylene blue) and electrode surface [186], thereby reducing interference from non-target species. Molecular recognition relies on complementary shape and charge interactions to distinguish structurally similar analytes, such as pathogenic bacteria [187,188] or ions [189].

Aptamer-based electrochemical sensors are broadly classified into labeled and label-free systems. Labeled sensors, such as redox-tagged aptasensors (e.g., ferrocene-labeled probes for *S. aureus* assay) or enzyme-linked platforms (e.g., HRP-conjugated aptamers for analyzing fusaric acid in cereal) [190], utilize covalent tags to modulate electrochemical signals. In contrast, label-free sensors rely on impedance changes (e.g., neomycin detection in milk samples) or electrochemiluminescence (ECL) (e.g., black hole quencher-based systems for arsenite quantification in rice grains) [191] to transduce binding events. Ratiometric sensors further improve accuracy by normalizing environmental noise through dual-signal systems, such as entropy-driven strand displacement reaction for kanamycin assay in food [192].

**Figure 7 foods-14-02669-f007:**
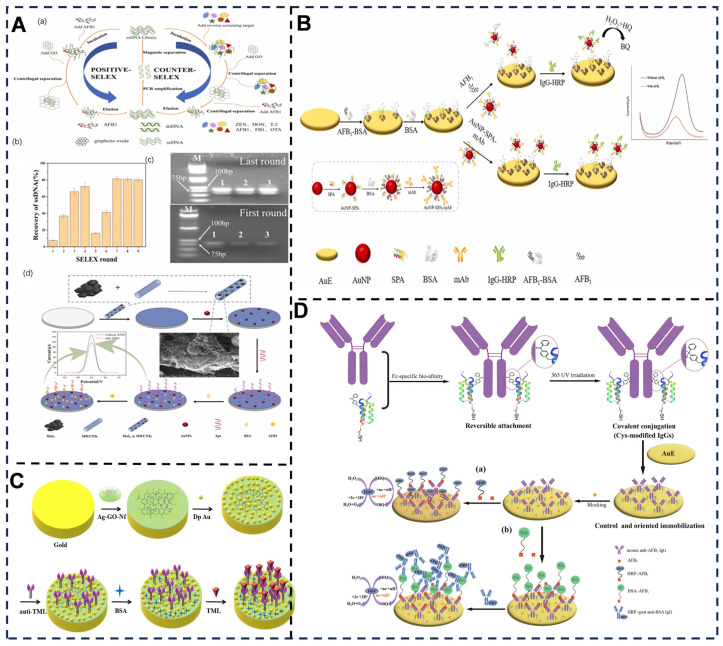
(**A**) (a) GO-SELEX screening for AFB1 aptamers. (b) ssDNA recovery rate per round. (c) Gel electrophoresis of initial/final PCR products (n = 3). (d) Aptasensor assembly [193]. (**B**) Immunosensor construction steps [194]. (**C**) TML immunosensor preparation process [195]. (**D**) Fabrication of Fc-Cys antibodies/immunosensors for AFB1 detection via competitive assays (direct/sandwich) [196]. Adapted with permission from Refs. [193,194,195,196]. Han et al. (2025), Xu et al. (2024), Wang et al. (2022) and Zhang et al. (2025).

### 4.2. Antibody-Enhanced Selectivity in Electrochemical Sensing

Antibodies, as immunoglobulin proteins with antigen-binding specificity [197,198], play a pivotal role in advancing the selectivity of electrochemical sensors (Figure 7B–D). Their Fab (antigen-binding fragment) regions enable precise molecular recognition of target analytes, effectively distinguishing structurally similar molecules even in complex matrices such as food samples. In electrochemical systems, antibodies are immobilized on electrode surfaces to form biorecognition layers. This immobilization leverages covalent bonding (e.g., via glutaraldehyde crosslinking) or physical adsorption, ensuring stable orientation of antigen-binding sites [199]. Upon antigen–antibody binding, measurable electrochemical signals, such as changes in current, impedance, or capacitance, are generated through redox mediators or label-free mechanisms [200].

Classification of antibody-based electrochemical sensors primarily depends on assay formats and antibody types. For instance, sandwich assays utilize dual antibodies targeting distinct epitopes of large antigens [201], amplifying specificity by requiring simultaneous binding events. Conversely, competitive assays employ labeled antigens to compete with analytes for limited antibody-binding sites, ideal for detecting small molecules like pesticides. Antibody classes also influence sensor design: IgG, with its high affinity and stability, dominates in commercial biosensors, while IgE-modified electrodes are tailored for allergen monitoring due to their hypersensitivity [202]. Similarly, nanobodies (single-domain antibodies) have been integrated into impedimetric sensors for real-time monitoring of cardiac biomarkers, achieving <5% cross-reactivity with interfering proteins [203]. Hybrid approaches combining IgM pentamers with CRISPR-Cas12a trans-cleavage [204] further improved selectivity in viral RNA detection, reducing false positives by 90% compared to conventional PCR. Aflatoxin B1 (AFB1) [205], the most toxic mycotoxin, has been addressed through two sensor strategies: label-free sensors based on a nanobody-modified black phosphorus/carbon nanotube interface achieve a detection limit of 0.27 pM, whereas labeled sensors utilizing HRP@MAF-7 biomimetic mineralization probes [206] further reduce the detection limit to 20 fg/mL; both strategies demonstrate recovery rates between 90.38% and 99.64%, meeting the EU’s strict standard of 2 μg/kg.

### 4.3. Cell-Enhanced Selectivity in Electrochemical Sensing

Cellular systems play a transformative role in advancing the selectivity of electrochemical sensors by leveraging intrinsic biological recognition mechanisms and dynamic metabolic responses [207] (Figure 8A–C). At the core of this synergy lies the ability of cells to interact with target analytes through membrane-bound receptors, ion channels, and enzymatic processes, which generate measurable electrochemical signals. For instance, neural stem cells or cancer cells [208] immobilized on electrode surfaces can selectively capture biomarkers via surface receptors, translating molecular recognition events into quantifiable current or impedance changes. This biohybrid approach capitalizes on the natural specificity of cellular components [209], such as antigen–antibody interactions or ligand-gated ion channels, to minimize cross-reactivity with interfering substances.

Classification of cellular electrochemical sensors primarily revolves around functional integration strategies [213]. Whole-cell biosensors utilize intact microorganisms or mammalian cells to detect toxins or nutrients through metabolic activity-modulated electron transfer, exemplified by Shewanella oneidensis-based systems monitoring heavy metal ions via extracellular respiration [214,215]. Subcellular fraction sensors employ isolated organelles (e.g., mitochondria) or membrane vesicles to target specific biochemical pathways, such as cytochrome c release detection during apoptosis [216]. Engineered cell-mimetic platforms integrate synthetic lipid bilayers embedded with purified cellular receptors, combining biological specificity with synthetic durability, a strategy demonstrated in dopamine sensing using reconstituted neuronal membrane proteins [217].

In the detection of complex food matrices, cell-enhanced electrochemical biosensors achieve highly selective detection through the specific response mechanisms of living cells: for biogenic amine contamination in aquatic products, genetically engineered yeast sensors utilize histamine-activated Ca^2+^ channel opening to induce impedance changes, enabling precise detection of 0.3 μM histamine [218]; for heavy metal contamination in grains and aquatic products, PC12 cells leverage Cd^2+^ blockage of K^+^ channels to alter membrane potential, generating a current response with a selectivity coefficient for Cd^2+^ over Pb^2+^ as high as 10.2 [219]; for pesticide residues in fruits and vegetables, *Chlorella* microalgae sensors operate on the principle of chlorpyrifos inhibiting photosynthetic electron transport, leading to decreased photocurrent and achieving recovery rates of 92.5–106.8% [220]; and for pathogenic bacterial toxins such as aflatoxin B1 in grain and oil products, human hepatoma cells (HepG2) generate impedance signal changes upon toxin-induced apoptosis, demonstrating a sensitivity of 0.05 ng/mL [221]. This cell physiology-based detection strategy significantly enhances the recognition accuracy of target analytes in complex food samples.

### 4.4. Protein Scaffold-Enhanced Selectivity in Electrochemical Sensing

Protein scaffolds have emerged as pivotal tools for enhancing the selectivity of electrochemical sensing platforms by leveraging their inherent molecular recognition capabilities, spatial organization, and modular adaptability [222] (Figure 8D). These biomolecular frameworks function as precision-engineered interfaces that immobilize recognition elements (e.g., enzymes, antibodies, aptamers) or catalytic moieties (e.g., nanoparticles, redox mediators) in optimal orientations, ensuring efficient target binding and signal transduction. The selectivity amplification arises from their ability to create microenvironmental niches that exclude interferents while concentrating analytes through affinity-driven interactions. For instance, antibody–antigen binding on protein scaffolds like protein A/G [223] ensures directional immobilization of antibodies, preserving their paratope accessibility and minimizing nonspecific adsorption [224]. Similarly, enzyme-loaded scaffolds, such as streptavidin–biotin systems, enable localized catalytic amplification; glucose oxidase (GOx) [225] immobilized on bovine serum albumin (BSA) scaffolds enhances electron transfer via direct wiring to electrodes, achieving selective glucose detection in serum with a limit of detection (LOD) of 2 μM.

Structurally, protein scaffolds are categorized into natural, engineered, and hybrid systems. Natural scaffolds, including ferritin cages and viral capsids (e.g., tobacco mosaic virus coat proteins), exploit their symmetric, porous architectures to encapsulate or display sensing components. Ferritin’s hollow core [226,227], for example, can house gold nanoparticles (AuNPs) for electrocatalytic detection of hydrogen peroxide [228], while viral capsids functionalized with DNA aptamers enable multiplexed detection of thrombin and ATP via spatially resolved binding sites. Engineered scaffolds, such as recombinant fusion proteins (e.g., SpyTag/SpyCatcher systems) and de novo designed peptides, offer programmable flexibility. The SpyTag/SpyCatcher covalent linkage system has been used to assemble cytochrome C and graphene oxide [229] into a stable biosensor for dopamine, achieving sub-nanomolar selectivity against ascorbic acid and uric acid [217]. Hybrid scaffolds integrate proteins with synthetic polymers or nanomaterials; for instance, DNA origami-protein [230] conjugates enable precise placement of redox reporters and aptamers, as demonstrated in a sensor for prostate-specific antigen (PSA) with a 10 pg/mL LOD.

Functional classification further highlights their roles as signal amplifiers, gatekeepers, or multifunctional integrators. Signal-amplifying scaffolds, such as horseradish peroxidase [231] (HRP)-loaded dendrimers, cascade enzymatic reactions to magnify currents. A tyrosinase-zeolitic imidazolate framework (ZIF-8) composite [232], for example, traps phenolic compounds within its pores, enabling ultrasensitive detection of bisphenol A at 0.1 nM. Gatekeeper scaffolds employ conformational changes to regulate analyte access; calmodulin-modified [233] electrodes undergo calcium-dependent structural shifts, selectively permitting glucose entry to GOx-active sites. Multifunctional scaffolds, like albumin-based matrices co-immobilizing capture antibodies and electroactive dyes, integrate recognition, enrichment, and reporting steps, as seen in a cortisol sensor for saliva analysis with 90% specificity [234].

Recent innovations underscore their versatility. Artificial protein scaffolds, designed via computational modeling [235] (e.g., ROSETTA-designed α-helical bundles), are tailored to bind heavy metal ions (e.g., Hg^2+^) through customized cysteine-rich pockets, achieving parts-per-trillion detection in wastewater. Stimuli-responsive scaffolds, such as elastin-like polypeptides (ELPs), undergo phase transitions under temperature or pH changes, dynamically tuning sensor surfaces to reject interferents. A pH-responsive ELP [236] scaffold loaded with laccase enabled selective detection of catechol in wine samples by reversibly exposing active sites only at acidic pH [237]. Challenges persist, including scaffold stability under operational conditions and scalability for industrial applications. Emerging solutions involve crosslinked protein–MOF hybrids, such as GOx-encapsulated ZIF-8, which enhance durability while maintaining enzymatic activity over 30 days.

In food safety monitoring, protein scaffold-enhanced electrochemical biosensors demonstrate critical applications across multiple hazard detection scenarios: For pathogen toxins (e.g., staphylococcal enterotoxin B in dairy and meat products), aptamer-functionalized scaffolds enable impedance-based detection with a 0.1 pg/mL limit of detection (LOD), bypassing culturing steps [238,239]; in pesticide residue analysis (such as chlorpyrifos in fruits and vegetables), acetylcholine esterase (AChE) anchored on nano-gold/carbon nanotube scaffolds quantifies organophosphates via inhibited thiocholine oxidation current within 0.01–100 μg/L range, achieving >95% recovery despite pigment interference [59]; for heavy metal monitoring (Cd^2+^/Hg^2+^ in aquatic products and grains), metallothionein-modified electrodes detect ion-binding induced current shifts by square wave voltammetry (SWV), reaching 0.05 ppb Cd^2+^ LOD with multi-metal capability; and in illegal additive screening (e.g., melamine in dairy), molecularly imprinted protein scaffolds provide specific recognition down to 0.1 ppm through differential pulse voltammetry (DPV) [240]. These applications leverage the molecular recognition properties of protein scaffolds to significantly overcome matrix interference in complex food samples while delivering rapid (≤30 min), sensitive (ppt–ppb level) detection.

### 4.5. MIP-Enhanced Selectivity in Electrochemical Sensing

Molecular imprinting technology (MIT) has revolutionized electrochemical sensing by enabling the creation of synthetic receptors with molecular-level specificity [241], mimicking the lock-and-key recognition mechanisms of food systems (Figure 9). The principle hinges on the formation of tailor-made cavities within a polymer matrix during polymerization, where template molecules (analytes) are embedded and later removed, leaving behind complementary binding sites that exhibit high affinity and selectivity for the target [242]. This process involves three stages: (1) pre-complexation of functional monomers (e.g., pesticide) with template molecules via covalent or non-covalent interactions [243]; (2) cross-linking polymerization with monomers such as ethylene glycol dimethacrylate (EGDMA) to stabilize the imprinted structure; and (3) template extraction, often using solvents or electrochemical etching, to expose the cavities. The resulting MIPs act as selective “artificial antibodies,” preferentially rebinding the target analyte even in complex matrices like blood, wastewater, or food extracts [244]. For instance, a dopamine-imprinted polypyrrole [245] film on a carbon electrode achieves sub-micromolar detection limits by selectively capturing dopamine through π-π stacking and hydrogen bonding while rejecting structurally similar interferents like ascorbic acid.

MIP-based electrochemical sensors are classified by imprinting strategy, material composition, and signal transduction mechanisms. Covalent imprinting [246], where templates are linked to monomers via reversible bonds (e.g., boronic acid–diol complexes), offers precise cavity geometry, as demonstrated in a glucose sensor using 4-vinylphenylboronic [247] acid to imprint glucose, achieving a 0.1 mM limit of detection (LOD) in serum. Non-covalent imprinting [248], relying on weaker interactions (hydrogen bonds, van der Waals forces), dominates practical applications due to its simplicity and versatility. A notable example is an ampicillin sensor [249] employing acrylic acid monomers to imprint the antibiotic via hydrogen bonding, enabling selective detection in milk at 5 nM. Hybrid imprinting combines covalent and non-covalent approaches for challenging targets; for example, a cortisol-imprinted sensor [234] uses a thiol-functionalized monomer for covalent anchoring and hydrophobic interactions to enhance binding specificity in sweat analysis [250]. Material-wise, MIPs integrate with conductive polymers (polyaniline, PEDOT), nanomaterials (graphene, MXenes), or MOFs to amplify signals. An MOF-MIP hybrid [251] (ZIF-8@MIP) for detecting bisphenol A leverages ZIF-8’s high surface area to increase cavity density, achieving a 0.01 nM LOD via differential pulse voltammetry.

Functional diversity is further illustrated through advanced MIP architectures. Electrochemically mediated imprinting allows real-time tuning of binding sites. A photoelectrochemical [252] sensor for ochratoxin A uses UV-induced polymerization on TiO_2_ nanotubes, creating light-responsive cavities that regenerate under irradiation. Dual-template MIPs enable multiplexed detection; a sensor co-imprinting caffeine and theophylline [253] on Au nanoparticle-decorated carbon dots distinguishes both analytes in coffee with 95% specificity.

In food safety analysis, MIPs significantly enhance the selectivity of electrochemical biosensors by serving as synthetic antibody-like recognition elements, enabling precise detection of contaminants in complex matrices. For mycotoxin monitoring [254] (e.g., aflatoxin B1 in grains), MIP-modified electrodes capture target molecules via shape-complementary cavities, generating redox current changes with ultra-low detection limits of 0.008 ng/mL; in pesticide residue [255] detection (such as chlorpyrifos in fruits), MIP-grafted quantum dots on sensor surfaces enable differential pulse voltammetry [256] (DPV) measurements through inhibited electron transfer, achieving 92–107% recovery across 0.1–50 μg/mL; for antibiotic [257] screening (e.g., enrofloxacin in animal-derived foods), MIP-based impedimetric sensors exhibit selective binding-induced capacitance shifts with 0.05 nM sensitivity and 10^3^ selectivity over structural analogs; and in illegal additive identification [258] (like bisphenol A in canned foods), MIP-carbon nanocomposite electrodes facilitate square-wave anodic stripping voltammetry (SWASV) detection at 0.05–0.1 ng/mL levels, resisting interference from saccharides and lipids. These MIP-engineered platforms collectively demonstrate rapid (<25 min), cost-effective, and matrix-tolerant analysis while maintaining stability under harsh food processing conditions.

**Figure 9 foods-14-02669-f009:**
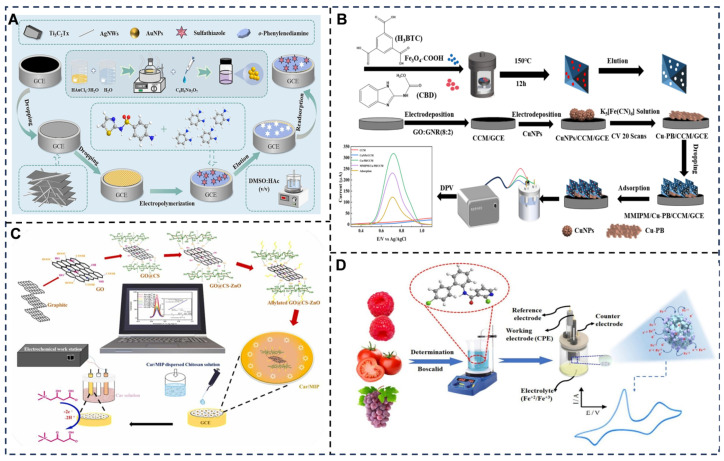
(**A**) Schematic preparation of MIP/AuNPs/AgNWs/Ti_3_C_2_Tx/GCE and electrochemical detection of STZ [259]. (**B**) Principle of construction and inspection of MMIPM/Cu-PB/CCM/GCE [260]. (**C**) An electrochemical nanosensor for levocarnitine (Car) detection using MIP as the molecular recognition platform [261]. (**D**) A novel BOS electrochemical sensing platform based on MOF-modified waste carbon conversion material, constructing a molecularly imprinted electrode structure with high sensitivity to BOS species [262]. Adapted with permission from Refs. [259,260,261,262]. Wang et al. (2025), Zhou et al. (2025), Athira et al. (2025) and Tarek et al. (2025).

## 5. Electrochemical Sensing Toward Food Testing

Benefiting from the obvious merits of rapid response, high sensitivity, and low cost, electrochemical sensing has emerged as a versatile and transformative tool in food analysis, addressing critical challenges in safety, quality, and authenticity. Its applications span three primary domains: contaminant detection [263], additive and nutrient analysis, and food authenticity verification, each supported by innovative sensing strategies and real-world implementations. Furthermore, electrochemical sensors possess multiplexing capability, thereby enabling simultaneous detection of multiple analytes.

### 5.1. Electrochemical Sensing Toward Food Contaminant

For contaminant detection, electrochemical biosensors excel in identifying trace-level hazards such as heavy metals, mycotoxins, and pathogens [87,264] (Figure 10). For instance, gold nanoparticle-modified electrodes paired with anodic stripping voltammetry enable the detection of lead ions (Pb^2+^) in drinking water at concentrations as low as 0.1 ppb, leveraging the pre-concentration of metal ions on the electrode surface [265]. Similarly, MIPs designed for aflatoxin B1 [266] in grains achieve detection limits of 0.05 ng/mL by measuring impedance changes caused by toxin binding, rivaling traditional chromatographic methods in sensitivity but with significantly reduced cost and time. Pathogen monitoring is enhanced by CRISPR-Cas13a-integrated platforms [178,267], where targeted bacterial DNA sequences trigger Cas13a’s collateral cleavage activity, releasing redox reporters like methylene blue [268] for ultrasensitive detection of the target gene of *S. aureus* as low as 2.4 copies mu L^−1^.

Electrochemical biosensors for food contaminant detection leverage tailored recognition interfaces and signal transduction strategies to achieve rapid, on-site analysis of complex matrices. For mycotoxins [254] (e.g., aflatoxin B1 in grains), aptamer-functionalized screen-printed electrodes generate redox current changes upon target binding with 0.01 ng/mL sensitivity, outperforming ELISA by 50-fold in peanut samples [269]; in pesticide residue monitoring (organophosphates in fruits), acetylcholinesterase-modified [270] nanoporous gold electrodes quantify chlorpyrifos via enzyme inhibition-induced thiocholine oxidation current reduction (linear range: 0.1–200 μg/L, recovery: 94.2–106.8%); for heavy metals [271] (Cd^2+^/Hg^2+^ in seafood), bismuth-film electrodes enable anodic stripping voltammetry detection at 0.05 ppb levels with <5% interference from coexisting ions; regarding veterinary drug residues [272] (e.g., enrofloxacin in milk), MIP-based sensors exhibit capacitance shifts through selective rebinding (LOD: 0.03 ng/mL, detection time: 15 min); and for pathogens [273] (Salmonella in meat), antibody-conjugated magnetic nanoparticles coupled with impedimetric sensors achieve 10 CFU/mL sensitivity without culture enrichment. These platforms demonstrate field-deployable capability (<30 min assay time), ppt–ppb level sensitivity, and >90% accuracy across diverse food systems, including oils, dairy, and produce.

**Figure 10 foods-14-02669-f010:**
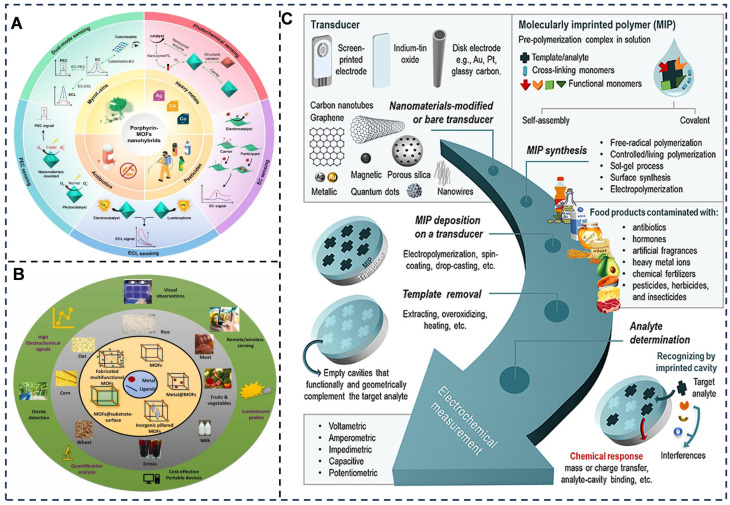
(**A**) Recent advances in photo/electrochemical biosensing of chemical food contaminants based on the porphyrin-MOFs nanohybrids [274]. (**B**) MOF-based luminescent and electrochemical sensors for food contaminant detection [275]. (**C**) Molecularly imprinted polymer-based electrochemical sensors for food contaminant determination [276]. Adapted with permission from Refs. [274,275]. Saleh et al. (2025) and Mohan et al. (2023).

### 5.2. Electrochemical Sensing Toward Additive and Nutritional Analysis

In additive and nutrient analysis, electrochemical sensors provide rapid, on-site quantification of sugars, preservatives, and antioxidants to ensure regulatory compliance and product consistency (Figure 11). Glucose oxidase-based biosensors remain a cornerstone for measuring sugar levels in beverages, where enzymatic oxidation of glucose generates a measurable current proportional to concentration [277]. For preservative analysis, polyphenol oxidase-modified electrodes detect sulfites in dried fruits by monitoring current inhibition caused by sulfite–enzyme interactions, achieving detection within 5 min [278]. Antioxidant levels in olive oil are quantified using laccase-encapsulated MOF sensors, which stabilize the enzyme and amplify electron transfer for precise measurement of phenolic compounds [279]. These systems bypass the need for complex sample pretreatment, even in opaque or viscous matrices like dairy products [280], by relying on electrical signals unaffected by optical interferences.

Electrochemical biosensors enable precise quantification of food additives and nutritional components by leveraging biomolecular recognition and nanomaterial-enhanced signal transduction, achieving rapid analysis in complex matrices. For preservatives (e.g., benzoic acid in beverages), molecularly imprinted polypyrrole electrodes detect oxidation current peaks at +0.82 V via square-wave voltammetry (SWV) with 0.07 μM LOD and 98.2–102.3% recovery [281]; regarding sweeteners (aspartame in dairy products), enzyme-modified Prussian blue/carbon nanotube sensors quantify hydrolysis-induced H_2_O_2_ reduction current (linear range: 0.5–500 μM, RSD < 3.5%) [282]; in antioxidant analysis (ascorbic acid in fruits), ZnO nanoflower-functionalized electrodes amplify electrocatalytic oxidation signals (sensitivity: 0.89 μA/μM, response time < 10 s); for micronutrients (vitamin D_3_ in fortified oils), aptamer-based impedimetric platforms exhibit target-binding capacitance shifts with 0.15 ng/mL LOD [283]; and concerning macronutrients (glucose in honey), glucose oxidase-immobilized 3D graphene aerogels achieve amperometric detection at 0.01–20 mM range (accuracy: 95.4–103.7% vs. HPLC) [284]. These systems demonstrate ≤15-min analysis, ppb-level sensitivity, and >94% accuracy across beverages, oils, dairy, and processed foods.

**Figure 11 foods-14-02669-f011:**
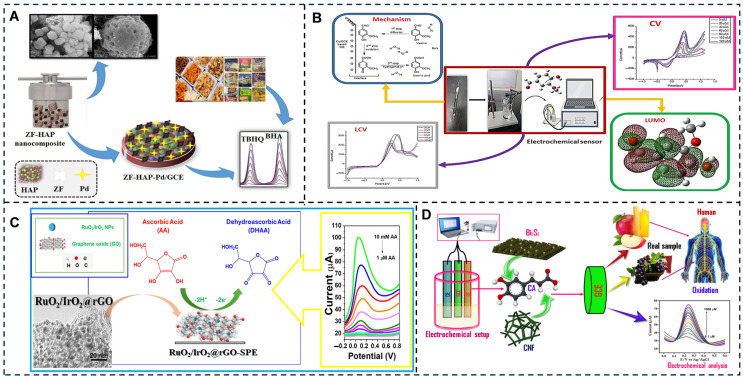
(**A**) Palladium–zinc ferrite-varnished hydroxyapatite spherocuboids for electrochemical detection of carcinogenic food preservatives [285]. (**B**) Metal nanocomposites-based electrochemical sensor for the detection of vanillin (food additives)—experimental and theoretical approach [286]. (**C**) RuO_2_/IrO_2_ nanoparticles on graphene-modified screen-printed electrode for enhanced electrochemical detection of ascorbic acid [287]. (**D**) Rapid detection of caffeic acid in food beverages using a non-enzymatic electrochemical sensor based on a Bi_2_S_3_/CNF nanocomposite [288]. Adapted with permission from Refs. [285,286,287]. Jyoti et al. (2025), Kumari et al. (2023) and Cirillo et al. (2023).

### 5.3. Electrochemical Sensing Toward Food Authenticity

Food authenticity verification is another critical application, combating fraud such as species substitution, geographic mislabeling, or adulteration (Figure 12). Electrochemical DNA sensors [289], for example, identify adulteration in premium products like salmon or truffle oil by hybridizing species-specific DNA probes with target sequences, with signal amplification via redox labels like ferrocene [289]. In olive oil authentication, MIP-based sensors selectively bind oleic acid, where binding-induced steric hindrance reduces electron transfer efficiency, distinguishing extra virgin oils from blended or oxidized counterparts [290]. Melamine detection in dairy products employs aptamer-functionalized electrodes, where melamine–aptamer binding alters interfacial capacitance, enabling quantification at 0.5 ppm—well below regulatory thresholds [291]. Emerging technologies like wearable sensors further enhance real-time monitoring; ethylene-sensitive polyaniline films [292] embedded in fruit packaging wirelessly transmit ripeness data, while self-powered lactose/O_2_ biofuel cells continuously monitor fermentation parameters in dairy plants without external energy sources.

Despite these advancements, challenges persist in standardization, multiplexed detection, and sensor durability. Future directions emphasize integrating artificial intelligence (AI) for data interpretation, such as neural networks trained on impedance spectra to authenticate honey botanical origins with >98% accuracy [293]. Blockchain-linked sensors [294] are being explored to immutably record real-time quality data (e.g., pesticide residues in produce) for supply chain transparency. Sustainable solutions like biodegradable chitosan-based sensors for phosphate detection in aquaculture align with global sustainability goals [295]. By harmonizing precision, scalability, and adaptability, electrochemical sensing is poised to redefine food safety ecosystems, ensuring compliance, reducing waste, and fostering consumer trust in an increasingly complex global food landscape.

Electrochemical biosensors for food authenticity assessment leverage species-specific biomolecular recognition and fingerprint-based signal profiling to combat adulteration in complex food matrices. For geographical origin verification (e.g., olive oil), DNA-based screen-printed electrodes detect region-specific genetic markers via hybridization-induced current changes (LOD: 0.1 fmol/L, discrimination accuracy > 98%) [296]; regarding species substitution (bovine milk in caprine products), aptamer-functionalized graphene electrodes quantify species-differential microRNAs through impedimetric shifts (0.01% *v*/*v* detection) [297]; in premium product fraud (saffron adulteration), MIP-coated quantum dot sensors generate voltammetric fingerprints of signature crocins (RSD < 2.5%, recovery: 96–104%); for processing method authentication (cold-pressed vs. refined oils), enzyme-modified nanoporous gold electrodes profile peroxide value kinetics (Δcurrent/min correlation: r^2^ = 0.997) [298]; and for ingredient dilution (honey adulteration with syrups), nanostructured zirconia electrodes detect maltooligosaccharide contaminants via catalytic oxidation peaks (specificity: 100% at 5% adulteration level). These platforms achieve ≤20-min analysis with <0.1% false-negative rates across oils, dairy, spices, and beverages.

**Figure 12 foods-14-02669-f012:**
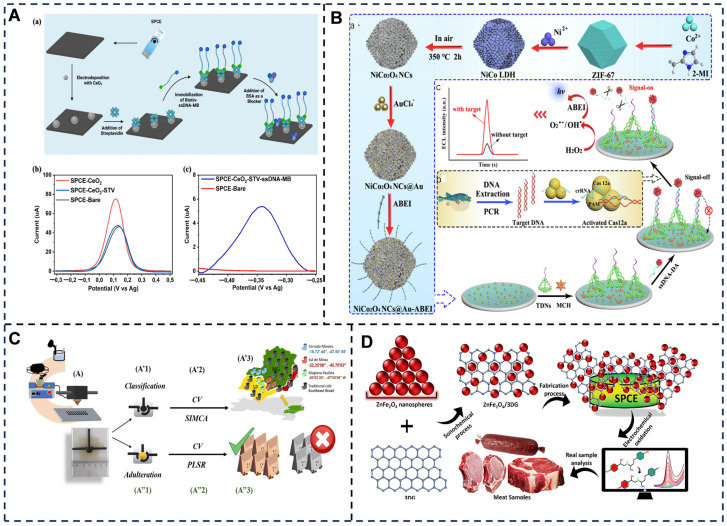
(**A**) SPCE modification steps and voltammetry: (a) electrode modification; (b) DPV: bare/ceria/STV-SPCE in [Fe(CN)_6_]^4−^/KCl; and (c) SWV: bare/ceria/STV/ssDNA-MB-SPCE in MB redox system [299]. (**B**) ECL biosensor: (a) NiCo_2_O_4_ NCs@Au-ABEI synthesis; (b) CRISPR/Cas12a target DNA amplification (PCR); and (c) biosensor fabrication/detection [137]. (**C**) Coffee authentication: (**A**) 3D-printed electrode fabrication; (A’) geographic classification: (A’1) Cell assembly → (A’2) CV/SIMCA/PLS-DA → (A’3) Regional classification; and (A”) blend quantification: (A”1) Au-modification → (A”2) CV/PLSR → (A”3) Adulteration verification [300]. (**D**) nM-level ractopamine detection in meat using ZnFe_2_O_4_/3D-graphene nanosheets for anti-fraud control [301]. Adapted with permission from Refs. [137,301]. Zhang et al. (2024) and Balram et al. (2024).

### 5.4. Multiplexing Capability of Electrochemical Biosensors

The multiplexing capability of electrochemical sensors enables simultaneous multi-analyte detection through four core strategies: spatial isolation, signal encoding, material enhancement, and catalytic amplification. Spatially isolated sensors employ microarray electrode designs, exemplified by dual-channel screen-printed carbon electrodes (SPCEs) modified with region-specific gold nanoparticle-carbon nanofiber (AuNPs/CNFs) composites [302]. This configuration independently detects lincomycin (LOD: 0.02 pg/mL) and neomycin (LOD: 0.035 pg/mL) with zero cross-talk across a six-order dynamic range (1.0 μM~0.1 pM). Signal-encoded systems leverage nanoscale electrochemical fingerprints, such as mixed-valence Ce(III,IV)-MOF synchronously catalyzing thionine reduction (−0.35 V peak for chlorpyrifos) and ferrocene reduction (+0.25 V peak for malathion), achieving sub-picomolar detection limits (0.038–0.045 pM) for dual pesticides [303]. Material-enhanced sensors utilize porous synergism, where MOF composites (Fe@YAU-101) exploit their ultrahigh surface area (>1200 m^2^/g) to concurrently capture Hg^2+^/Pb^2+^/Cd^2+^ with detection limits surpassing 10^−10^ M (Cd^2+^: 6.67 × 10^−10^ M) [304]. Similarly, BiCu alloy-carbon core–shell structures (BiCu-ANPs@CF) prevent structural deformation through carbon encapsulation, enabling tri-ionic detection of Pb^2+^ (0.081 ppb), Cd^2+^ (0.95 ppb), and Zn^2+^ (35 ppb) across broad ranges (0.5–900 ppb) [305]. Catalytically amplified systems integrate entropy-driven catalysis (EDC) with magnetic separation, as demonstrated by Fe@ZIF67-NH_2_ ratiometric sensors that independently amplify Pb^2+^/Hg^2+^ signals while reducing background noise by 80% via magnetic isolation of carbon-dot-labeled DNA strands (CDP-D1/CDH-P1), achieving ultra-low detection limits (0.03–0.1 ng/mL) [306]. All platforms exhibit exceptional reproducibility (RSD < 6.6%) and reliability in real-sample validation (recovery rates: 96.4–107.0%).

Multiplexed sensors demonstrate revolutionary potential in environmental and health monitoring: Their high efficiency (e.g., 16-channel electrodes screening antibiotics/toxins/pathogens/heavy metals within 15 min) and low cost (SPEs < USD 0.5/unit—99% cheaper than mass spectrometry) [307] overcome limitations of conventional methods (e.g., HPLC/immunoassays) in operational complexity and cross-reactivity. Successful deployments include on-site analysis of seafood heavy metals, human biofluids (Pb^2+^/Cd^2+^ recovery > 95%), and environmental water (trace antibiotic monitoring) [308]. Nevertheless, challenges persist in crosstalk suppression (probe interference > 3%), biorecognition stability (<30-day shelf life at ambient temperature), and standardization gaps. Future advancements will focus on three frontiers: artificial intelligence-driven voltammetric deconvolution (CNN models boosting multi-signal discrimination accuracy > 97%) [309], microfluidic-nanopore integration for “sample-to-answer” detection (e.g., 8-analyte monitoring within 100-μm^2^ zones), and blockchain-embedded data traceability (e.g., real-time uploads to IBM Food Trust) [310]. These innovations will catalyze a paradigm shift from “single-analyte ultra-sensitivity” to “multi-target intelligent early-warning” systems, ultimately establishing real-time surveillance networks for environmental pollution screening, food safety control, and early disease diagnostics.

## 6. Current Achievements and Future Horizons

### 6.1. State of the Art in Electrochemical Biosensors

Electrochemical sensing has cemented its role as a transformative technology in food analysis, offering unparalleled advantages in speed, sensitivity, and adaptability for monitoring food composition, safety, and quality [311,312]. The evolution of electrochemical platforms has been driven by innovations in materials science, microfabrication, and data analytics. Nanostructured materials, including MOFs and MXenes [313], enhance sensor performance by increasing active surface areas and catalytic activity. An MOF-encapsulated laccase biosensor [314], for example, achieves ultrasensitive detection of phenolic antioxidants in olive oil by stabilizing the enzyme and amplifying electron transfer. Wearable and wireless sensors are reshaping on-site monitoring; flexible epidermal patches with lactate and pH electrodes track meat spoilage in real time during refrigerated transport, while smart packaging embedded with ethylene-sensitive polyaniline films wirelessly alerts suppliers to fruit ripeness. Lab-on-a-chip (LOC) systems integrate microfluidics [314] with multiplexed electrochemical arrays to simultaneously profile multiple analytes, such as nitrates, sulfites, and ascorbic acid in processed foods, using machine learning to deconvolute overlapping signals. Despite these advancements, challenges persist in standardizing sensors for heterogeneous food matrices, mitigating biofouling in lipid-rich samples, and ensuring long-term stability under variable storage conditions.

Electrochemical biosensing technology still faces multiple technical bottlenecks in food analysis—the primary challenge stems from the extreme complexity of food matrices (components such as lipids, proteins, and pigments readily adsorb irreversibly onto electrode surfaces, causing signal drift or baseline distortion/signal attenuation, exemplified by oil film interference during benzo [a]pyrene detection in edible oils; notably, protein/lipid coverage (e.g., olive oil) [315] on active sites leads to 38% attenuation in Pb^2+^ signal (optical sensors show only 8% attenuation), while electroactive substances like ascorbic acid induce potential shifts (e.g., 0.16 V offset in rutin detection in fruit juice); and high-salt environments (e.g., Na^+^/Cl^−^ in seawater) suppress enzymatic activity [316], causing a 4-fold response delay in microbial sensors, and pH fluctuations (e.g., lemon juice pH ≈ 2.5) reduce dehydrogenase NAD^+^ conversion efficiency by >60%). Simultaneously, biorecognition elements (especially natural enzymes and antibodies) exhibit significantly degraded stability under high temperatures and extreme pH in processing environments (natural enzymes have a half-life < 2 h at 60 °C; tyrosinase suffers > 30% sensitivity decay after 10 cycles due to polyphenol byproducts blocking active sites, and NAD^+^-dependent dehydrogenase systems typically have ambient storage periods under 30 days) [317], with their ambient storage periods generally shorter than 1 month (except for aptamers, which can exceed 6 months), severely constraining on-site deployment feasibility. Furthermore, practical samples like meat and juice require cumbersome pretreatment (enzymatic hydrolysis to remove proteins or centrifugation to separate solids), extending detection procedures (meat detection process prolonged by 50%), while batch-to-batch variations in nanomaterial synthesis (particle size fluctuations ± 20%; causing signal fluctuations ± 25%) and unstable antibody immobilization efficiency result in insufficient data reproducibility (e.g., carbon film cracking in BiCu alloy-carbon core–shell structures after repeated heavy metal adsorption reduces electroactive area by 25%, and insulating layers formed by food lipids cause 40% current decay in GaCu nanozymes), hindering industrial translation [318]. More critically, structural analogs of target analytes (e.g., sulfonamide antibiotics) often induce cross-reactivity, and the current lack of unified validation standards casts doubt on the comparability of detection limits across studies.

Long-term stability challenges are primarily constrained by the dual mechanisms of biorecognition element deactivation and electrode material degradation: electrode materials face structural collapse risks (e.g., carbon film cracking in BiCu-ANPs@CF after repeated adsorption), and insulating layers from food lipids exacerbate electrode passivation [319]. To overcome this bottleneck, cutting-edge research focuses on biomimetic protection technologies and core–shell structure design—polydopamine (PDA) encapsulation maintains > 90% activity in microbial sensors after 15 tests; β-cyclodextrin-embedded NAD^+^ combined with polyvinylpyrrolidone film extends test strip shelf life to 6 months; carbon film-encapsulated BiCu alloy improves cycling stability 5-fold by inhibiting volumetric deformation, while the disposable design of screen-printed electrodes (SPE) (unit cost < USD 0.5) fundamentally circumvents aging issues. Strategies against matrix interference integrate physical separation and signal correction technologies. Magnetic separation-assisted entropy-driven catalysis (EDC) systems reduce background noise by 80% through capturing carbon dot-labeled DNA chains (CDP-D1/CDH-P1) [320], achieving a 0.03 ng/mL detection limit for Hg^2+^ in seafood; MOF nanoporous membranes (ZIF-8, pore size 0.34 nm) selectively filter proteins, enhancing flavonoid detection selectivity in juice by >100-fold; and Fe@ZIF67-NH_2_ bifunctional materials serve as internal references for real-time pH fluctuation correction, yielding RSD < 4% in meat detection. Future efforts require establishing accelerated aging test standards (ISO 17025, 40 °C/90% RH) and developing self-healing hydrogel materials to achieve >1-year cold-chain monitoring lifespan; microfluidic pretreatment chips will integrate lipid separation and pH adjustment within 5 min, while AI-driven calibration (CNN deconvolution of overlapping peaks) and EC-P dual-mode sensing can eliminate 99% of false positives, ultimately constructing an intelligent detection paradigm integrating anti-interference and self-correction capabilities [321]. Additionally, we compare the strengths and limitations of electrochemical versus optical biosensors in Table 2.

### 6.2. Future Perspectives

Looking ahead, the convergence of electrochemical sensing with emerging technologies promises to address current limitations and unlock novel applications. To overcome the constraints of electrochemical biosensing technology, research is now focusing on strategies such as anti-fouling coatings (e.g., zwitterionic polymers to shield nonspecific adsorption; ccaerogels to resist lipid adsorption) [326], biomimetic recognition elements (molecularly imprinted polymers replacing natural antibodies; artificial nucleic acid enzymes resistant to 100 °C high temperatures), and microfluidic chip-integrated pretreatment modules (enabling 15-min “sample-in-result-out”). Additionally, artificial intelligence algorithms are being explored for automatic real-time signal calibration and compensation, which will play a pivotal role in optimizing sensor design and data interpretation; neural networks trained on impedance spectra of honey samples can distinguish botanical origins and adulteration patterns with >98% accuracy. Concurrently, smart system upgrades (microbial fuel cells for self-powering [113]; enzymatic fuel cells that harvest energy from analytes like glucose or lactose) could enable autonomous, continuous monitoring in industrial fermentation tanks or agricultural fields, and ecosystem development (ISO/AOAC cross-platform certification, cloud database sharing) will accelerate industrialization. Blockchain-integrated sensors are poised to revolutionize traceability, where real-time electrochemical data on pesticide residues or allergen levels in perishables is immutably recorded, enabling automated compliance verification across global supply chains and empowering disruptive applications such as blockchain-based full-chain data notarization and in vivo antibiotic metabolism tracking in livestock. Furthermore, the rise of biodegradable sensors (e.g., chitosan-based electrodes for phosphate detection in aquaculture) aligns with sustainability goals. Multi-omics integration, combining electrochemical data with metabolomic, proteomic, or genomic profiles, will enhance predictive modeling of food quality and safety—for instance, correlating electrochemical signals of lipid oxidation markers (e.g., malondialdehyde) with metabolomic shifts in fried snacks may forecast shelf-life stability and flavor degradation [327].

In conclusion, electrochemical sensing stands at the forefront of a paradigm shift in food analysis, bridging laboratory-grade precision with field-deployable practicality. Future research must prioritize scalability, interoperability, and regulatory harmonization to translate lab innovations into industry-standard tools. By synergizing advancements in nanotechnology, synthetic biology, and digital twins, electrochemical systems will ultimately propel this technology from discrete devices toward an intelligent food safety network—not only safeguarding food integrity but also empowering personalized nutrition, smart agriculture, and circular food economies, unlocking trillion-dollar market potential.

## Figures and Tables

**Figure 1 foods-14-02669-f001:**
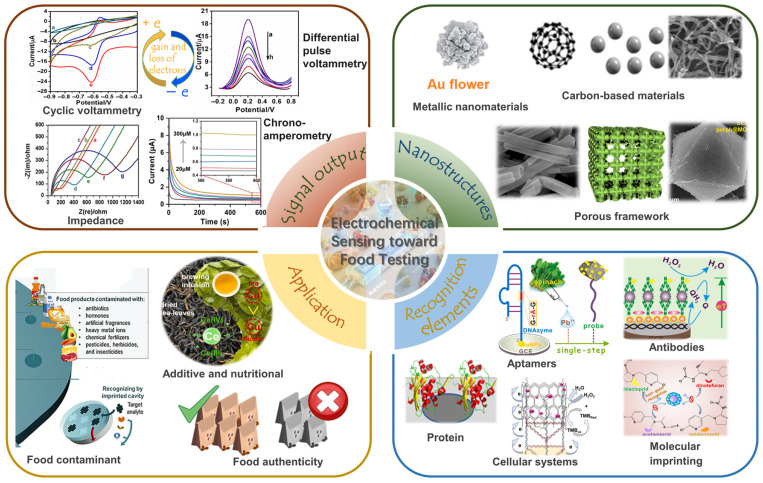
Schematic diagram of electrochemical sensing toward food testing.

**Figure 2 foods-14-02669-f002:**
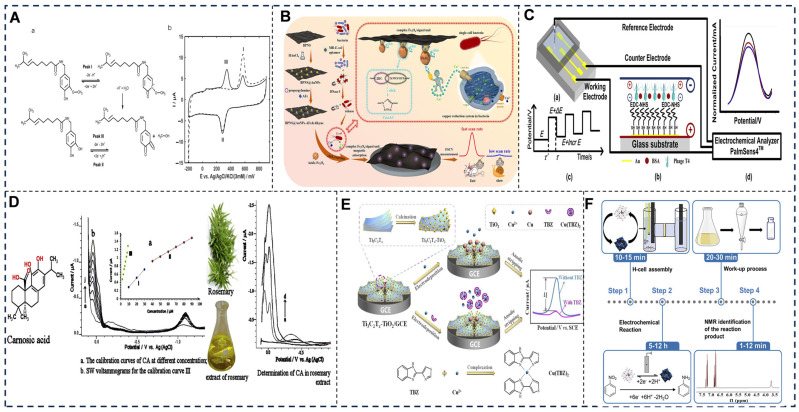
(**A**) Adsorptive stripping voltammetry: (a) reaction pathway; (b) 1st/2nd scan voltammograms [92]. (**B**) Click chemistry-triggered electrochemical sensor with fast scan voltammetry [93]. (**C**) *E. coli* detection via T4B-MES/DPV: (a) 3-microelectrode sensor; (b) electric field-assisted T4 phage immobilization; and (c,d) DPV measurements (Pulse: 1 ms/0.5 V, Step: 7 mV) [94]. (**D**) Direct SWV quantification of carnosic acid [95]. (**E**) Ti_3_C_2_T_x_-TiO_2_/GCE preparation and sensing mechanism [96]. (**F**) Redox-mediated chronoamperometric reduction in nitrobenzene [97]. Adapted with permission from Refs. [92,93,94,96]. Søpstad et al. (2019), Zhou et al. (2023), Xu et al. (2020) and Zhong et al. (2023).

**Figure 3 foods-14-02669-f003:**
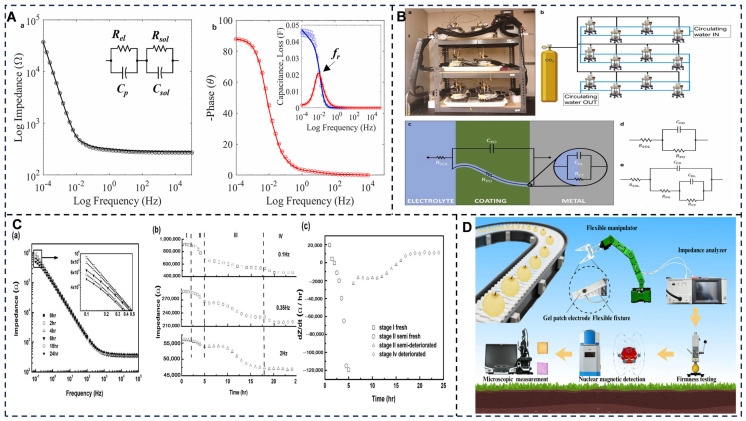
(**A**) EIS of PU/PEDOT:PSS electrodes in BG11 (14 days): (a) |Z| vs. freq. (0.1 mHz–100 kHz) w/equivalent circuit (CpǁRel–Rsol); (b) Phase vs. freq. w/Cp and loss tangent (Gp/ω) [106]. (**B**) Multi-chamber EIS setup: (a,b) Experimental design; (c–e) Equivalent circuits modeling electrolyte penetration (initial → substrate stages [107]. (**C**) Impedance evolution during storage: (a) |Z| vs. freq.; (b) |Z| vs. time; (c) Rate of change vs. time [108]. (**D**) Nondestructive EIS maturity assessment of Nanguo pears using gel-patch electrodes + flexible gripper, correlated with microscopy/NMR/firmness/physicochemical analyses [109]. Adapted with permission from Refs. [107,109]. Filippas et al. (2024) and Sui et al. (2025).

**Figure 4 foods-14-02669-f004:**
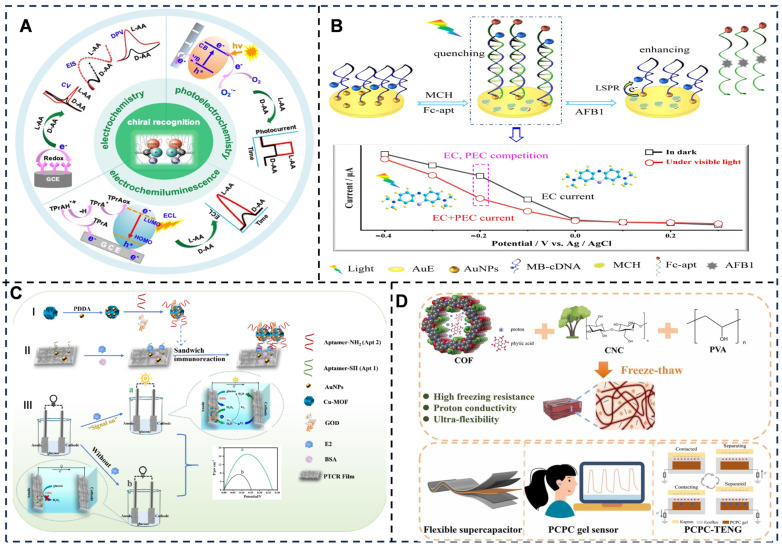
(**A**) Illustration of the chiral electrochemical, electrochemiluminescent, and photoelectrochemical-based methods [115]. (**B**) illustrations of AFB1 detection based on the electronic competition between PEC-driven and EC-driven redox reactions of MB modulated by plasmonic AuNPs [114]. (**C**) Schematic diagrams of (I) the construction process of Cu-MOF@AuNPs/GOD/Apt2, (II) the construction process of the Cu-MOF@AuNPs/GOD/Apt2-E2-Apt1 sandwich structure, and (III) the working principle of the self-powered sandwich-type aptamer sensor for the detection of E2 [116]. (**D**) Ultra-flexible anti-freezing cellulose conductive hydrogel for energy storage and self-powered sensors [117]. Adapted with permission from Refs. [114,115,116,117]. Pan et al. (2024), Li et al. (2022), Ning et al. (2024) and Jia et al. (2025).

**Figure 5 foods-14-02669-f005:**
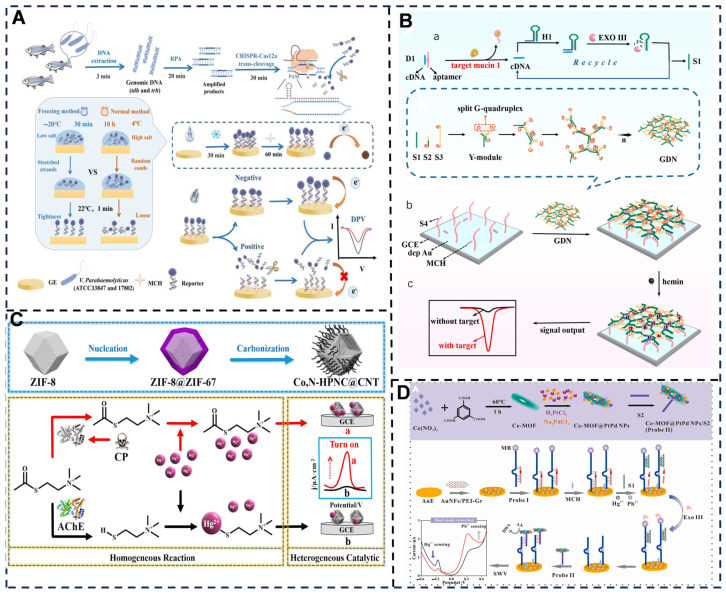
(**A**) Schematic illustration of the RPA-mediated E-CRISPR electrochemical biosensor for detection of Vp [131]. (**B**) (a) Exo III-Assisted Target Mucin 1 Recycling Amplification; (b) preparation of the G-quadruplex-Enriched DNA Nanonetwork; and (c) construction of Electrochemical Biosensor for the Detection of Target Mucin 1 [132]. (**C**) Schematic diagram of the preparation process of Co,N-HPNC@CNT and the detection principle of this sensor [133]. (**D**) The preparation process of the signal label (Probe II); schematic diagram of a sensor for simultaneous detection of Hg^2+^ and Pb^2+^ [134]. Adapted with permission from Refs. [131,132,133,134]. Xu et al. (2025), Wang et al. (2025), Liu et al. (2024) and Hui et al. (2024).

**Figure 8 foods-14-02669-f008:**
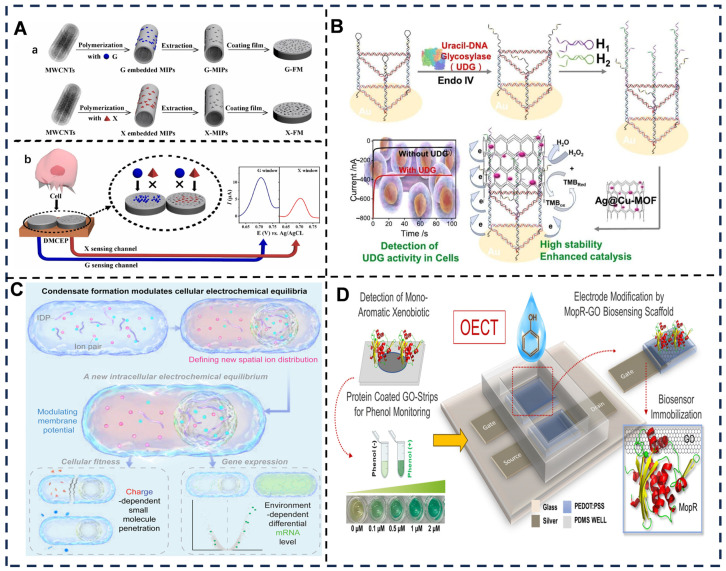
(**A**) (a) Preparation of G-FM/X-FM; (b) DMCEP enables simultaneous intracellular G/X detection [210]. (**B**) The Ag@MOF electrochemical biosensor uses a nano-DNA prism to enhance catalysis, achieving high-sensitivity UDG activity detection in cellular systems [207]. (**C**) Biomolecular condensates modulate bacterial electrochemical environments (cytoplasmic pH, membrane potential), influencing global cellular processes and amplifying intercellular variability [211]. (**D**) MopR-OECT sensor for rapid, ultrasensitive phenol detection (LOD: 2 ppb) [212]. Adapted with permission from Refs. [207,210,211]. Zhang et al. (2023), Cong et al. (2025) and Dai et al. (2024).

**Table 1 foods-14-02669-t001:** Electrochemical biosensors for food detection applications.

Category	Key Content	Technical Highlights and Performance Parameters
Detection Targets		
Small Molecules	Toxins (aflatoxin B1), additives (benzoic acid/aspartame), and pesticides (organophosphates)	MIPs for organophosphates: Recovery > 95% Aptamer for AFB1: K_d_ ≈ 10^−9^ M
Microorganisms	Pathogens (*Salmonella* and *E. coli*) and viruses	CRISPR-Cas visualization: False positive rate < 0.5% Response time < 20 min
Ions	Heavy metals (Pb^2+^ and Hg^2+^) and trace elements	DNAzyme for Hg^2+^: LOD 0.1 ppt Interference resistance > 90%
Macromolecules	Allergenic proteins, DNA adulteration markers	Nanopore sensors: Species identification accuracy > 98%
Recognition Elements		
Aptamers	SELEX-selected DNA/RNA and conformational switching for small molecules	Reusability > 50 cycles and cost reduction 80% Label-free design capability
Antibodies	Sandwich/competitive assays for pathogens/toxins	Affinity constant K_A_ ≈ 10^10^ M^−1^
Molecularly Imprinted Polymers (MIPs)	Synthetic cavities for pesticides/additives	Tolerance: organic solvents/pH 2–12 Template recovery > 95%
Cells/Enzymes	Whole-cell metabolic monitoring (histamine) and enzymatic catalysis (glucose oxidase)	Biomimetic recognition and suitability for in vivo analysis
Detection Principles		
Direct Current (DC)	Voltammetry (DPV/SWV): heavy metals/additives; Amperometry: steady-state catalytic current	Sensitivity: 10^−8^–10^−12^ M Wide linear range (3 orders)
Alternating Current (AC)	EIS for interface changes (e.g., bacterial adhesion); Conductometric sensors	Label-free real-time monitoring LOD: 10^2^–10^3^ CFU/mL
Emerging Modes	Photoelectrochemistry (PEC): light-controlled e^−^ transfer; Self-powered systems: microcystin-RR detection	Background interference resistance External power-free operation
Signal Amplification		
Nanomaterials	Au/Ag NPs, graphene, and MOFs for enhanced surface area	AuNPs-SERS: LOD 10^−18^ M Electron transfer efficiency × 5
Catalytic Cascades	Enzyme-linked reactions (HRP), nanozymes (Fe_3_O_4_@MIL-101), and CRISPR-Cas amplification	Signal gain > 100× Single-base mismatch specificity
Redox Cycling	[Ru(NH_3_)_6_]^3+^/mediator pairs	Background noise suppression > 90%
Application Scenarios		
Contaminant Screening	Mycotoxins (AFB1), heavy metals, and pathogens	On-site detection: < 30 min Sensitivity: ppt–ppb level
Nutrients and Additives	Antioxidants (ascorbic acid), preservatives, and sweeteners	Eliminates sample pretreatment (oils/dairy) RSD < 5%
Authenticity Verification	DNA sensors for meat adulteration; MIPs for olive oil markers	Geographical origin accuracy > 98% Processing method identification
Future Directions		
Intelligent Systems	AI-assisted EIS analysis (neural networks); Blockchain data traceability	Reduced human error Supply chain transparency
Sustainability	Biodegradable electrodes (chitosan/cellulose); Microbial fuel cells	Environmental compatibility Green chemistry alignment
System Integration	Microfluidic chip coupling; Flexible wearable sensors (patch-type detection)	Portable on-site deployment Real-time continuous monitoring
Standardization Challenges	Lack of unified performance criteria; Matrix interference in real samples	Requires ISO/ASTM standards development

DPV: differential pulse voltammetry; EIS: electrochemical impedance spectroscopy; MIPs: molecularly imprinted polymers; AuNPs: gold nanoparticles; MOFs: metal–organic frameworks; RSD: relative standard deviation.

**Table 2 foods-14-02669-t002:** Comparison of electrochemical vs. optical biosensors in food testing.

Parameter	Electrochemical Biosensors	Optical Biosensors	Technical Impact	SERS
Detection Limit	0.01–0.1 ppb (Nanomaterial amplification)	0.1–10 ppb (SERS/Fluorescence)	Electrochemical biosensors superior for trace contaminants	[322]
Response Time	<10 min (Direct electron transfer)	≥30 min (SPR equilibrium required)	Electrochemical biosensors ideal for on-site decisions	[322]
Equipment Cost	USD 50–USD 500 (Handheld reader)	USD 5000–USD 50,000 (SPR/Raman systems)	Electrochemical cost: 1–10% of optical	[323]
Multiplexing Capability	Excellent (16-channel microarray)	Moderate (3-channel fluorescence)	Electrochemical biosensors enable high-throughput	[305,324]
Matrix Interference Resistance	Poor >30% signal attenuation (e.g., olive oil)	Excellent (SPR resists nonspecific adsorption)	Optical biosensors are better for dark/turbid samples (e.g., juice)	[322]
Long-term Stability	<30 days (Enzyme activity decay)	>6 months (Fiber-optic sensors)	Optical biosensors suitable for long-term monitoring	[323]
Spatial Resolution	None	Micrometer-level (Fluorescence imaging)	Optical biosensors uniquely visualize contaminant distribution	[325]
Sample Pretreatment Demand	High (Enzymatic digestion for meat)	Low (SERS direct detection of turbid liquids)	Optical biosnesors reduce processing by 50%	[322]
Typical Applications	-Pesticide screenin -Cold chain monitoring	-Food fraud lab verification -Non-destructive packaged food testing	Complementarity > Competition	[323]

## Data Availability

No new data were created or analyzed in this study. Data sharing is not applicable to this article.

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
