# Peer review of "Electrochemical Biosensors Driving Model Transformation for Food Testing"

_foods, 2025, doi:10.3390/foods14152669_

Round 1
Reviewer 1 Report
Comments and Suggestions for Authors
The paper by Wu et al. on the electrochemical biosensors for food testing reviews the recent literature for the use of electrochemical biosensors, their operational mechanisms, emphasizing nano-enhanced signal amplification, and biorecognition elements. The topic of review is very interesting. My main comment is that there is no introduction to how biosensors work or an explanation of their advantages and disadvantages in food analysis. Simply listing a large number of articles in which biosensors have been used (387) seems to me to be of little use. On the other hand, there is a lack of a summary table to navigate through the review article's content more easily for the reader. A discussion section should be added, include a collective summary of insights from the authors regarding the advantages, disadvantages and future perspectives of electrochemical biosensor development for food safety application.
Reviewer 2 Report
Comments and Suggestions for Authors
The review article titled "Electrochemical Biosensors Driving Model Transformation for Food Testing" explores the transformative role of electrochemical biosensors in ensuring food safety and quality control. In contrast to conventional methods, which are often limited by high cost, operational complexity, and poor applicability in field settings, electrochemical biosensors offer rapid analysis, high sensitivity, and the ability to operate without extensive sample pretreatment. The article comprehensively discusses signal amplification strategies (e.g., nanostructures, enzymatic catalysis), biorecognition elements (aptamers, antibodies, and molecularly imprinted polymers), and practical applications including contaminant detection, nutrient analysis, and food fraud prevention. Furthermore, emerging trends such as artificial intelligence, biodegradable sensors, and blockchain integration are addressed, contributing to the development of a sustainable and intelligent food safety ecosystem.
-
The importance of food safety and the limitations of existing methods could be more clearly articulated, particularly through a more detailed comparison with optical-based techniques.
-
The multiplexing capabilities of electrochemical sensors, i.e., the simultaneous detection of multiple analytes, should be explored in greater detail, accompanied by practical examples.
-
Repetitive descriptions regarding signal amplification and the benefits of nanomaterials should be minimized to improve clarity and conciseness.
-
The references need to be revised in accordance with the journal’s formatting style.
-
Practical limitations, such as the long-term stability of the sensors and their performance in complex food matrices, should be more prominently emphasized.
Round 2
Reviewer 1 Report
Comments and Suggestions for Authors
The authors have made revisions to the article as suggested. The manuscript has met the requirements of the journal and is recommended for acceptance.